# SPINT1 Expressed in Epithelial Cells of Choroid Plexus in Human and Mouse Brains

**DOI:** 10.3390/ijms26115130

**Published:** 2025-05-27

**Authors:** Ryuta Murakami, Yoichi Chiba, Genta Takebayashi, Keiji Wakamatsu, Yumi Miyai, Koichi Matsumoto, Naoya Uemura, Ken Yanase, Yuichi Ogino, Masaki Ueno

**Affiliations:** 1Department of Pathology and Host Defense, Faculty of Medicine, Kagawa University, 1750-1 Ikenobe, Miki-cho, Kita-gun, Takamatsu 761-0793, Kagawa, Japan; murakami.ryuta@kagawa-u.ac.jp (R.M.); s20d727@kagawa-u.ac.jp (K.W.); miyai.yumi@kagawa-u.ac.jp (Y.M.); matsumoto.koichi@kagawa-u.ac.jp (K.M.); 2Department of Anesthesiology, Faculty of Medicine, Kagawa University, 1750-1 Ikenobe, Miki-cho, Kita-gun, Takamatsu 761-0793, Kagawa, Japan; s22d718@kagawa-u.ac.jp (G.T.); uemura.naoya@kagawa-u.ac.jp (N.U.); yanase.ken.a4@kagawa-u.ac.jp (K.Y.); ogino.yuichi@kagawa-u.ac.jp (Y.O.)

**Keywords:** choroid plexus, E-cadherin, epithelial cells, SIP1, SPINT1

## Abstract

The functional significance of the choroid plexus (CP), such as in the control of circadian rhythm as well as production of cerebrospinal fluid, is attracting attention. Transepithelial and junctional transport between epithelial cells of CP plays an important role in its function. Recently, an epithelial cadherin, E-cadherin, as well as non-epithelial cadherins were confirmed to be expressed in CP epithelial cells. The serine protease inhibitor Kunitz type 1 (SPINT1) is expressed in many kinds of epithelial cells and affects epithelial developmental function by controlling E-cadherin expression. However, it has not been confirmed whether SPINT1 is expressed in epithelial cells of CP. Thus, in this study, we examined whether SPINT1 is expressed in CP epithelial cells by immunohistochemistry and RT-PCR. Immunohistochemical expression of SPINT1 was noted in the cytoplasm of epithelial cells of humans and mice, and mRNA of SPINT1 was expressed in samples derived from the CP of mice. SPINT1 was typically expressed in CP epithelial cells with E-cadherin and Smad-interacting protein (SIP1), an E-cadherin transcriptional repressor. Some enlarged epithelial cells showed strong SPINT1 signals. These findings indicate that SPINT1 is expressed in epithelial cells of CP in relation to E-cadherin expression.

## 1. Introduction

It is well known that the choroid plexus (CP) epithelium (CPE) plays a central role in the secretion of cerebrospinal fluid (CSF) from the blood to ventricles by transport of electrolytes and nutritional substances, such as glucose and lactate [1,2]. Water channels and electrolyte transporters in plasma membranes of epithelial cells and the junctions between them play significant roles in CSF production [3,4,5], suggesting the significant contribution of abnormalities in junctional components as well as epithelial cells to CP dysfunction. CP regulates behavioral circadian rhythms via junctional components [6]. The circadian rhythm can be disrupted by aging [7] and neurodegenerative conditions [8]. Thus, the known significance of junctional components in CP is increasing. Recently, E-cadherin was also reported to be expressed in the CPE of humans and mice [9]. Conversely, hepatocyte growth factor activator inhibitor type 1 (HAI-1)/serine protease inhibitor Kunitz type 1 (SPINT1) (HAI-1/SPINT1) is a membrane-associated serine proteinase inhibitor expressed in the cytoplasm of various epithelial tissues, such as the gastrointestinal tract, prostate, lung, and skin [10,11]. HAI-1/SPINT1 was also reported to be localized with E-cadherin in placental cytotrophoblasts [12,13]. The localization of HAI-1/SPINT1 in proliferating trophoblastic stem cells (cytotrophoblasts) suggests the possible role of HAI-1/SPINT1 in their proliferation. Thus, HAI-1/SPINT1 is critical for epithelial developmental function, possibly through regulation of cell-surface proteases. Cheng et al. [14] reported that HAI-1/SPINT1 knockdown in a human pancreatic cancer cell line (SUIT2) significantly reduced the expression of E-cadherin and was accompanied by increased expression of Smad-interacting protein 1 (SIP1), which is known to be an E-cadherin transcriptional repressor. Moreover, a metastatic variant of SUIT2 with loss of E-cadherin expression (S2-CP8) showed a significantly reduced level of HAI-1/SPINT1. Overexpression of HAI-1/SPINT1 in the cell line S2-CP8 resulted in increased E-cadherin and down-regulated SIP1 expressions. These findings suggest that changes in E-cadherin expression depend on or are related to changes in SPINT1 expression in carcinoma cell lines. However, SPINT1 has not been reported to be expressed in epithelial cells of CP. Thus, we initially examined whether SPINT1 was expressed in CPE by RT-PCR as well as immunohistochemical analysis in mice and humans. Secondly, we examined whether SPINT1 expression is related to the expression of E-cadherin or SIP1, an E-cadherin transcriptional repressor, in CPE of human brains.

## 2. Results

### 2.1. SPINT1 Expression in Mouse Choroid Plexus

In the mouse CP, immunoreactivities using a rabbit polyclonal antibody for SPINT1 (GTX114793) were noted in the cytoplasm and cytoplasmic membrane of epithelial cells (Figure 1A–C). RT-PCR analysis revealed that *Spint1* mRNA is expressed at almost the same level in CP tissues derived from the lateral and fourth ventricles, whereas the expression of mRNA of SPINT1, *spint1*, in CP tissues was lower than that in small intestinal tissue (Figure 1D). Sequence analysis of amplicons confirmed that they were as the expected fragments of mouse *spint1*.

### 2.2. SPINT1 Expression in Human Choroid Plexus

#### 2.2.1. Classical Immunohistochemical SPINT1 Expression

In the human CP, immunoreactivities using a monoclonal antibody for SPINT1 (sc-137159) in all ten brains were observed in the cytoplasm of epithelial cells (Figure 2). There was no clear difference in localization of immunoreactivities for SPINT1 among the ten brain samples.

#### 2.2.2. Double Immunofluorescence Examination of SPINT1 Expression

Double immunofluorescence examination using human samples showed that immunoreactivity for SPINT1 (sc-137159) was usually localized in the cytoplasm of CPE showing immunoreactivity for E-cadherin on the lateral membrane of CPE in a brain sample (Figure 3A–F). Immunoreactivity for E-cadherin was often decreased in the lateral cytoplasmic membrane of enlarged epithelial cells showing marked expression of SPINT1 (Figure 3D–F). In another sample, immunoreactivity for SPINT1 was usually localized in the cytoplasm of CPE showing immunoreactivity for E-cadherin on the lateral membrane of CPE, whereas SPINT1 immunoreactivity was sometimes inversely correlated with E-cadherin expression (Figure 3G–I). Immunoreactivity for SPINT1 was typically localized in the cytoplasm of CPE showing immunoreactivity for SIP1. Moreover, immunoreactivity for SIP1 was occasionally decreased in enlarged epithelial cells showing marked expression of SPINT1 and increased in epithelial cells showing weak SPINT1 expression (Figure 3J–O). Original unprocessed confocal microscopic images are shown in Appendix A. In addition, a double immunohistochemical examination using antibodies for SPINT1 and mitochondria or endosomes was conducted. Immunoreactivity for SPINT1 was not colocalized with that for mitochondria or endosomes (Appendix A).

#### 2.2.3. Morphometrical Analysis

Figure 4 presents the correlation between areas and brown ratios or DAB intensity in 4608 cells. An analysis of areas and brown ratios revealed a weak positive correlation between them (r = 0.359, *p* < 0.01) (Figure 4A). An analysis of areas and DAB intensity revealed a weak correlation between them (r = 0.298, *p* < 0.01) (Figure 4B). In addition, the correlation between the brown ratio and DAB intensity is shown in Appendix A. An analysis of the brown ratio and DAB intensity revealed a strong positive correlation between them (r = 0.836, *p* < 0.01).

## 3. Discussion

In mice, immunoreactivity for SPINT1 was noted in the cytoplasm and cytoplasmic membrane of CPE in the lateral ventricle, whereas mRNA expression of SPINT1, *spint1*, was detected in CP in the lateral and fourth ventricles. In human brains, immunoreactivity for SPINT1 was observed in the cytoplasm of CPE in the lateral ventricle of all autopsied human brains examined. These findings indicate that SPINT1 is expressed in CPE of mice and humans. Conversely, SPINT1 expression in the rat CP can be noted in the transcriptomic webserver database of the rat CP [15] “https://cprnaseq.in.ku.dk/ (accessed on 11 March 2025)”. In addition, immunoreactivity for SPINT1 was typically localized in the cytoplasm of epithelial cells showing immunoreactivity for E-cadherin or SIP1. There were slight individual differences in immunoreactivities among the 10 patients. In addition, E-cadherin or SIP1 immunoreactivity was sometimes decreased in enlarged epithelial cells showing marked immunoreactivity of SPINT1, whereas immunoreactivities for E-cadherin and SIP1 were occasionally increased in epithelial cells showing weak SPINT1 immunoreactivity. Morphometric analysis suggested a weak correlation between SPINT1 immunoreactivity and the cell size. Although cells smaller than those with a 200-pixel area were excluded, the possibility that spindle cells other than epithelial cells were very rarely stained and included in the evaluation could not be fully ruled out.

SPINT1/HAI-1 is known to be expressed on the surface of epithelial cells, and immunohistochemical studies have demonstrated membranous immunoreactivity in normal epithelial cells. Conversely, Oberst et al. [16] reported that the subcellular localization of immunohistochemical staining for SPINT1/HAI-1 was observed both in the cytoplasm and at the cell membrane in human breast carcinoma cells. They reported that the former may be explained by the internalization of proteins or the synthetic pool of these molecules. We examined double immunohistochemical investigations using antibodies for SPINT1 and mitochondria or endosomes. However, immunoreactivity for SPINT1 was not co-localized with that for mitochondria or endosomes. Some epithelial cells with enlarged cytoplasm showed strong immunoreactivity for mitochondria (Appendix A). Further examinations are needed to identify specific subcellular locations.

According to previous findings [14], SPINT1 expression strongly affects E-cadherin expression in association with the expression of SIP1, an E-cadherin transcriptional repressor, in specific carcinoma cell lines. In addition, SPINT1 regulated epithelial to mesenchymal transition (EMT) via interaction with serine proteinases in the cell line. A recent study showed that some epithelial cells exhibited strong expression of E-cadherin, whereas other cells showed weak or little expression of E-cadherin [9]. However, SPINT1 was reported to be required for maintenance of intestinal epithelial integrity [17]. Kataoka et al. [10] reported that regenerating epithelium in injured areas displayed increased expression of HAI-1/SPINT1 in comparison with epithelium in normal areas. Accordingly, strong immunoreactivity for SPINT1 in CPE of human brains noted in this study might be necessary for survival of enlarged epithelial cells which may be damaged. The enlarged epithelial cells were reported to be localized in CP of aged brains with neurodegenerative diseases [18]. The CP with damaged epithelial cells may affect the appearance and progression of abnormal neurological symptoms. However, the hypothesis was not supported by other experiments, such as in vitro experiments. Further experiments to confirm the roles of SPINT1 expression in CPE are warranted.

## 4. Materials and Methods

### 4.1. Human Tissues

Human tissue samples were obtained at autopsy in Kagawa University Hospital, as previously described [9,19]. Table 1 summarizes the clinical profiles of all subjects including the postmortem delay. We could remove ten brains within 12 hours after death at autopsy and detected several kinds of molecules including transporters [9,19,20], indicating that the brain materials were suitable for analysis of these molecules in this study. This study conformed to the Declaration of Helsinki and was approved by the Institutional Ethics Committee of the Faculty of Medicine, Kagawa University. After fixation with 10% formalin, paraffin-embedded tissue blocks were prepared and cut into 4 μm-thick sections.

### 4.2. Animals

Animal studies were approved by the Kagawa University Animal Care and Use Committee, and all efforts were made to minimize the number and extent of suffering of animals used. Under deep anesthesia with intraperitoneal injection of a medetomidine-midazolam-butorphanol mixture (0.3/4.0/5.0 mg/kg), 10-week-old male C3H/HeSlc mice (*n* = 6) (Japan SLC, Hamamatsu, Japan) were transcardially perfused with phosphate-buffered saline (PBS), followed by the procedures described below for immunohistochemical and RT-PCR analyses.

### 4.3. Immunohistochemistry

For immunohistochemical analysis of human CP, paraffin sections from the human medial temporal lobe including CP of autopsied human brains were deparaffinized and pretreated with 0.3% hydrogen peroxide in PBS for 30 min to block endogenous peroxidase activity. After blocking with 2% bovine serum albumin (BSA) in PBS for 30 min, sections were incubated with several kinds of antibodies at 4 °C overnight. Primary antibodies against SPINT1 (HAI-1) (1:500, rabbit polyclonal, Cat. No. GTX114793; Gene Tex) [21], SPINT1 (HAI-1) (1:200, mouse monoclonal, sc-137159, clone H-1; Santa Cruz Biotechnology, Dallas, TX, USA) [13], E-cadherin (1:2000, rabbit polyclonal, Cat. No. 20874-1-AP; ProteinTech Group, Chicago, IL, USA) [22,23], Smad-interacting protein 1 (SIP1) (1:200, rabbit polyclonal, C10133, Assay Biotechnology Company, Fremont, CA, USA), mitochondria (1:200, mouse monoclonal, 909-301-D79, Rockland, Limerick, PA, USA), and EEA1, a marker for endosomes (1:100, rabbit polyclonal, NBP1-30914, Novus, Centennial, CO, USA) [24] were used in this study. Prior to incubation with some antibodies, antigen retrieval was performed by heating sections in 10 mM sodium citrate buffer (pH 6) or 1 mM Tris-ethylenediaminetetraacetic acid (EDTA) (pH 9) at 95 °C for 20 min. After treatment with hydrogen peroxide and blocking with 2% BSA in PBS for 30 min, sections were incubated with primary antibodies at 4 °C overnight. They were then washed with PBS, incubated with a polymer solution conjugated with anti-rabbit IgG antibody or anti-mouse IgG antibody and horseradish peroxidase (HRP) (Histofine^®^ Simple Stain™ MAX PO (R), Nichirei Biosciences Inc., Tokyo, Japan), and developed with 3,3′-diaminobenzidine. The sections were counterstained with hematoxylin.

For double immunofluorescence examination, sections were incubated at 4 °C overnight with the mouse antibody for SPINT1 (1:200, sc-137159) and rabbit antibody for E-cadherin (1:200, 20874-1-AP) or Smad-interacting protein 1 (SIP1) (1:200, C10133), followed by incubation at RT for 60 min with Alexa Fluor 488-anti-mouse IgG (1:200, Abcam, ab150113) and Alexa Fluor 594-conjugated anti-rabbit IgG (1:200, Abcam, ab150080) antibodies, respectively. In addition, we performed double immunofluorescence examination using two-kinds of antibodies for SPINT1 (GTX114793) and mitochondria and those for SPINT1 (sc-137159) and EEA1, a marker for endosomes (NBP1-30914). Before incubation with primary antibodies, antigen retrieval was performed by heating sections in 10 mM sodium citrate buffer (pH 6) or 1 mM Tris-EDTA (pH 9) at 95 °C for 20 min. The sections were then incubated for 60 min at RT in Monomeric Cyanine Nucleic Acid Stain (TO-PRO-3, Molecular Probes, Eugene, OR, USA), which was diluted to 2.5 μM in PBS. The fluorescent signals were viewed under a confocal microscope (Carl Zeiss LSM700, Oberkochen, Germany). As a control experiment, we performed an identical immunohistochemical procedure with omission of the primary antibodies. For immunohistochemical analysis of mouse CP, mice (*n* = 3) were perfused with PBS, followed by perfusion with 4% paraformaldehyde in 0.1 M phosphate buffer (pH 7.4). Dissected tissues were postfixed with the same fixative at 4 °C overnight, embedded in paraffin, and cut into 4 μm-thick sections. Immunohistochemical staining was performed in the same way as for human brains.

### 4.4. Morphometry

Immunohistochemical staining for SPINT1 (sc-137159) in ten human brains was visualized, and images were captured using an Olympus microscope (Olympus, Tokyo, Japan) equipped with a 2.8-megapixel digital camera at a resolution of 1920 × 1440 pixels and ×400 total magnification. Ten randomly selected, non-overlapping microscopic fields were analyzed from each section. In cases where the CP tissue area was insufficient to obtain ten unduplicated views, all available CP tissue regions within the section were analyzed. Prior to image analysis, each captured image was resized to 30% of its original dimensions to optimize computational performance. Epithelial cells of CP were then automatically segmented and identified using Cellpose, a deep-learning-based cell segmentation algorithm. The expression level of SPINT1 in individual cells was evaluated by digitally quantifying the signals and calculating the ratio of the immunoreactive (brown-stained) area to total segmented area, which was defined as the brown ratio, in each epithelial cell. Image processing and quantitative analyses were conducted using custom Python scripts (version, 3.12.5) incorporating Cellpose, OpenCV, NumPy, and Pandas libraries, adapting methods described in previous papers [9,13]. Cells with a less than 200-pixel area, corresponding to cells less than 7 μm in diameter, were excluded to minimize non-epithelial cell contamination. In addition, DAB intensity was quantified as the mean pixel value of the V-channel in the HSV color space (range: 0–255). A total of 4608 cells were analyzed across all samples. The associations between cell areas and immunoreactive area ratios (brown ratios) and between cell areas and DAB intensity were statistically assessed using Spearman’s rank correlation coefficient. Significance was defined as a p-value less than 0.05.

### 4.5. RT-PCR

After perfusion with PBS, CP tissues in the lateral and fourth ventricles and small intestine tissues were isolated from mice (*n* = 3) and total RNA was extracted using ReliaPrep^™^ RNA Tissue Miniprep System (Promega, Fitchburg, WI, USA) [19]. cDNA was synthesized using ReverTra Ace^®^ qPCR RT Master Mix (Toyobo). Ten nanograms of cDNA were used as a template and amplified with PrimeSTARR Max DNA Polymerase (TAKARA BIO INC., Kusatsu, Japan) and specific primers [19]. Primer sequences used for RT-PCR were as follows: *spint1* sense 5′-CTGCGCAGGTCACAAGCA-3′, antisense 5′-GAGCGAACTTGCACACGAAG-3; *gapdh* sense 5′-CAAGGTCATCCATGACAACTTTG-3′, antisense 5’-GTCCACCACCCTGTTGCTGTAG-3’. PCR amplification cycle conditions were 25 or 28 (*gapdh*) and 33 or 35 (*spint1*) cycles of 10 s at 98 °C, 5 s at 55 °C, and 5 s at 72 °C. Amplicons were electrophoresed on a 2% agarose gel, stained with Midori Green Advance (Nippon Genetics, Tokyo, Japan), and visualized with blue LED (470 nm) transilluminator (AMZ System Science, Osaka, Japan). Amplified fragments were subjected to direct sequencing (Eurofins Genomics, Tokyo, Japan).

## 5. Conclusions

In mice and humans, SPINT1 is expressed in epithelial cells of the CP in relation to E-cadherin expression.

## Figures and Tables

**Figure 1 ijms-26-05130-f001:**
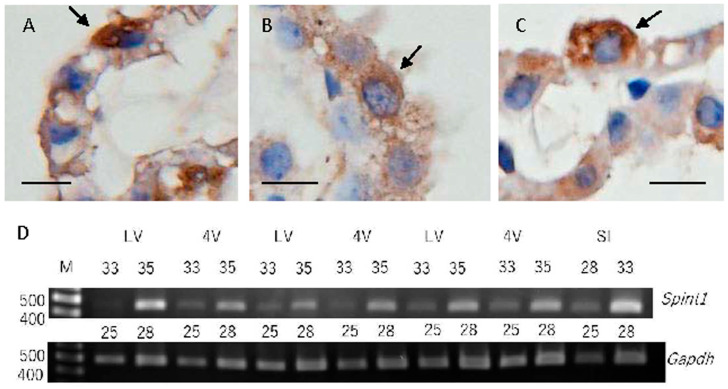
Representative microphotographs of immunoreactivities for SPINT1 (GTX114793) in mice (**A**–**C**) and expression of *Spint1* and *Gapdh* mRNAs in CP located in the lateral (LV) and fourth (4V) ventricles and small intestine (SI) (**D**). Arrows indicate clear or strong immunoreactivity for SPINT1 (**A**–**C**). Scale bars indicate 10 μm.

**Figure 2 ijms-26-05130-f002:**
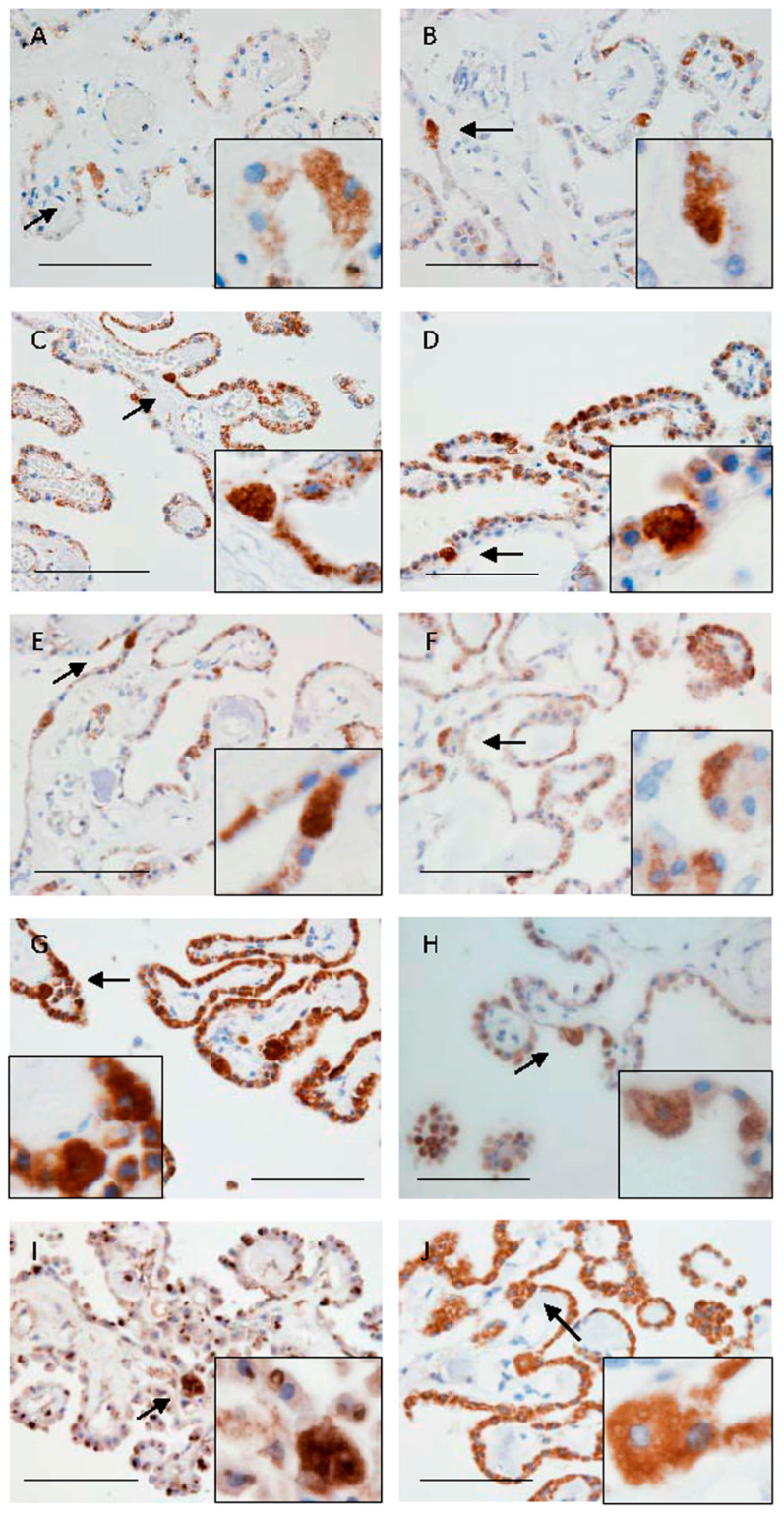
Representative microphotographs of immunoreactivities for SPINT1 (sc-137159) in humans (**A**–**J**). (**A**–**J**) are staining images for SPINT1 in No. 1–10 shown in Table 1, respectively. Arrows indicate clear or strong immunoreactivity for SPINT1 in epithelial cells with enlarged cytoplasm (**A**–**J**). Insets of (**A**–**J**) show strong immunoreactivities in the enlarged cells. Scale bars indicate 100 μm.

**Figure 3 ijms-26-05130-f003:**
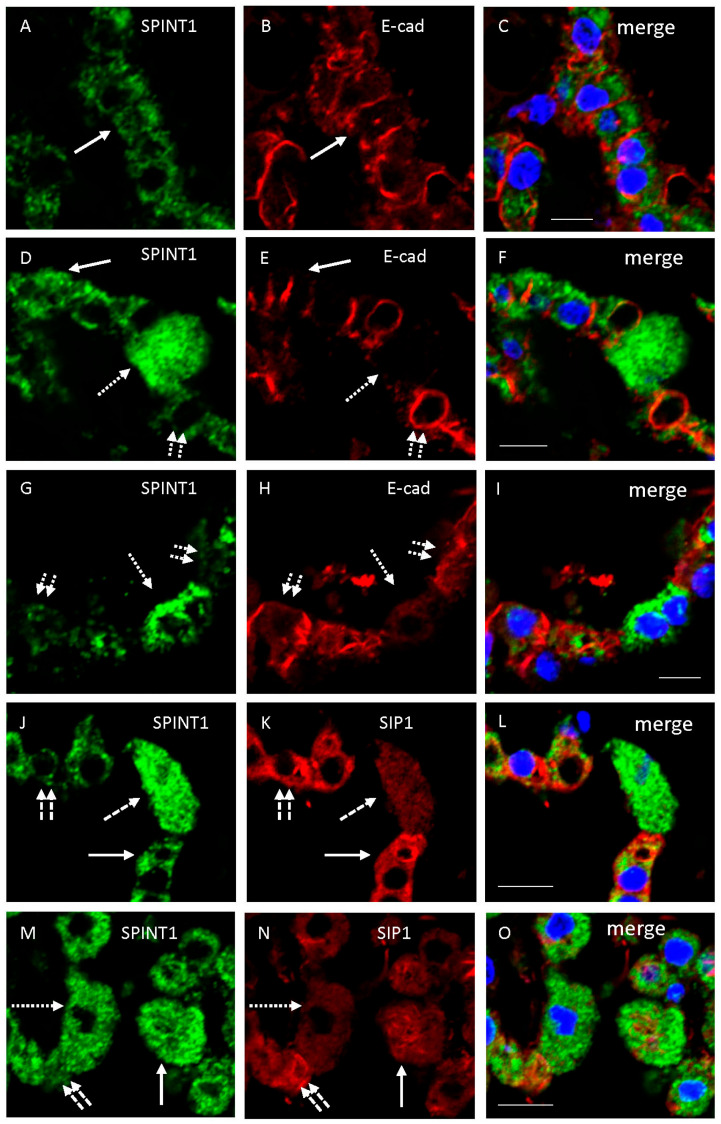
Representative confocal microscopic images of epithelial cells of CP from cases No. 7 (**A**–**F**,**J**–**L**), 9 (**G**–**I**), and 10 (**M**–**O**). Immunostaining with the mouse anti-SPINT1 antibody (**A**,**D**,**G**,**J**,**M**) (visualized as green), rabbit anti-E-cadherin (**B**,**E**,**H**) or SIP1 (**K**,**N**) antibody (visualized as red), nuclear staining by TO-PRO-3 (visualized as blue), and merged images (**C**,**F**,**I**,**L**,**O**) are shown. Solid long arrows in images (**A**,**B**,**D**,**E**,**J**,**K**,**M**,**N**) show double immunoreactivity. Dotted long arrows present strong SPINT1 and weak E-cadherin or SIP1 expression (**D**,**E**,**G**,**H**,**J**,**K**,**M**,**N**), whereas dotted double short arrows show weak SPINT1 and strong E-cadherin or SIP1 expression (**D**,**E**,**G**,**H**,**J**,**K**,**M**,**N**). Scale bars indicate 10 μm.

**Figure 4 ijms-26-05130-f004:**
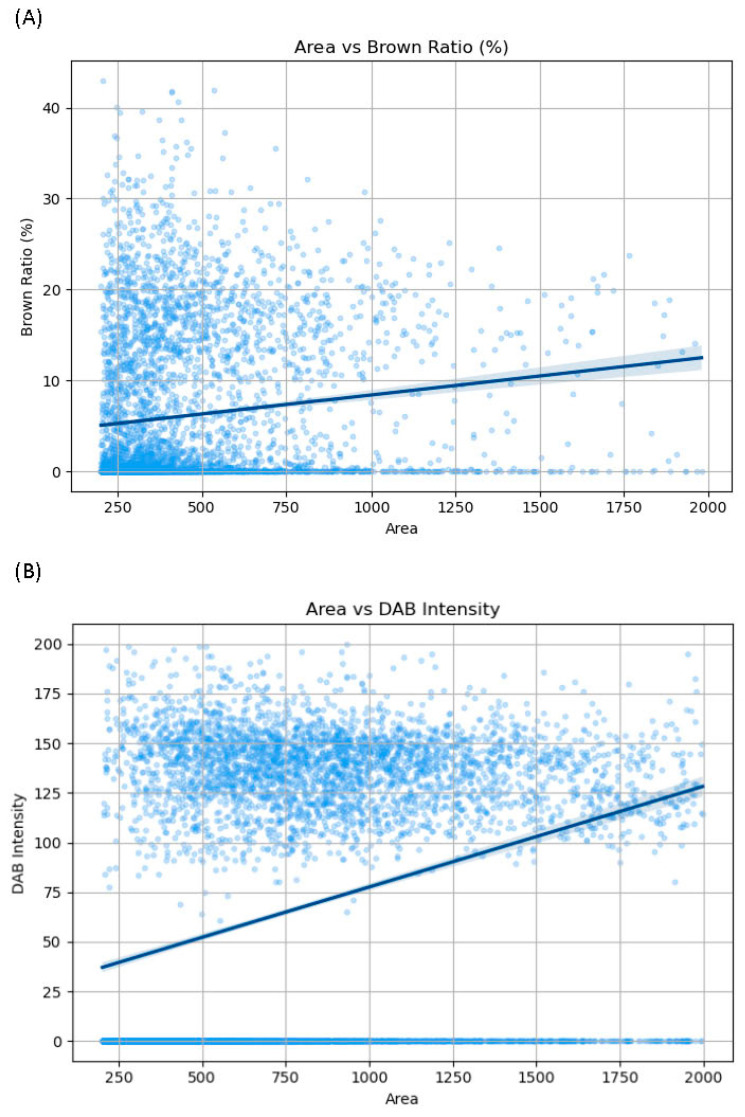
Scatterplots show the relationship (**A**) between the cell area and immunoreactive area ratio (brown ratio), defined as the percentage of the immunoreactive area relative to total cell area, and the relationship (**B**) between the cell area and DAB intensity, defined as the mean pixel value of the V-channel in the HSV color space (range: 0–255), for 4608 cells obtained from 10 brain samples. The horizontal axis represents the cell area measured in pixels, whereas the vertical axis shows the immunoreactive area ratio (brown ratio) (**A**) and DAB intensity (**B**). The area of one pixel is approximately 0.245 μm^2^.

**Table 1 ijms-26-05130-t001:** Summary of clinicopathological profiles.

(No.)	Age/Sex	Main Diagnosis	PMD (h)
1	42/M	Pulmonary hypertension, Heart failure	10
2	57/F	Psychiatric disorder, Liver abscess, Sepsis	12
3	64/M	Multiple system atrophy, Pneumonia	5
4	68/F	Thalamic hemorrhage	5
5	70/M	Myocardial infarction	2
6	72/F	Pneumonia	4
7	74/M	Lung cancer	10
8	75/M	Gastric cancer	1.5
9	75/M	Multiple system atrophy, Pneumonia	1
10	84/M	Myocardial infarction, Cerebral infarction	2

F, female; h, hour(s); M, male; PMD, postmortem delay.

## Data Availability

The original contributions presented in this study are included in the article. Further inquiries can be directed to the corresponding authors (M.U. and Y.C.).

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
