# Peer review of "SPINT1 Expressed in Epithelial Cells of Choroid Plexus in Human and Mouse Brains"

_ijms, 2025, doi:10.3390/ijms26115130_

Round 1

Reviewer 1 Report

Comments and Suggestions for Authors

In this short paper, the authors investigated the expression of SPINT1 (HAI-1) in choroid plexus (CP) epithelial cells using immunohistochemistry and RT-PCR, as well as its relationship to E-cadherin and SIP1 (an E-cadherin repressor). They found that SPINT1/HAI-1 is expressed in the cytoplasm of CP epithelial cells in both humans and mice and confirmed the presence of its mRNA in CP-derived samples from mice. Additionally, SPINT1/HAI-1 expression was sometimes inversely correlated with E-cadherin expression.

Major comments

  1. The data presented in this paper are convincing. However, the manuscript is entirely descriptive, leaving the reader with questions unanswered satisfactorily. While the expression of SPINT1/HAI-1 is a novel finding, no process has been explored in sufficient depth to provide insight into the roles and functions of this protein in CP cells. Additionally, no statistical analysis has been conducted to examine the relationship between the expression levels of SPINT1/HAI-1 and E-cadherin or SIP1. Furthermore, the connection between E-cadherin and SIP1 remains unclear.
  2. PINT1/HAI-1 is known to be expressed on the surface of epithelial cells, and immunohistochemical studies have demonstrated membranous immunoreactivity in normal epithelial cells. Therefore, the cytoplasmic expression lacking membranous immunoreactivity of SPINT1/HAI-1 in CP cells might be an interesting and novel finding. The authors should further investigate or at least discuss the biological significance of the cytoplasmic SPINT1/HAI-1 in CP cells.

Minor comment: Line 134: SPINT-1 should be SPINT1.

Author Response

To Reviewer 1:

Major comment (1):

The data presented in this paper are convincing. However, the manuscript is entirely descriptive, leaving the reader with questions unanswered satisfactorily. While the expression of SPINT1/HAI-1 is a novel finding, no process has been explored in sufficient depth to provide insight into the roles and functions of this protein in CP cells. Additionally, no statistical analysis has been conducted to examine the relationship between the expression levels of SPINT1/HAI-1 and E-cadherin or SIP1. Furthermore, the connection between E-cadherin and SIP1 remains unclear.

To major comment (1): Thank you for your reasonable comment. As shown in Fig. 3, SPINT1 was co-expressed in epithelial cells showing the expression of E-cadherin or SIP1. In addition, we found that epithelial cells with enlarged cytoplasm were in the choroid plexus. The cells frequently showed strong SPINT1 immunoreactivity and weak or no immunoreactivity of E-cadherin or SIP1. In addition, epithelial cells rarely showed weak SPINT1 immunoreactivity and clear E-cadherin or SIP1 immunoreactivity. It was impossible to show the exact frequency using double immunostaining with a confocal microscope. We think that it is meaningful to show the existence of enlarged epithelial cells showing increased immunoreactivity for SPINT1. On the other hand, double immunostaining using antibodies for E-cadherin and SIP1 did not work. We additionally examined the correlation between areas and DAB intensity and the correlation between brown ratio and DAB intensity and did a double immunohistochemical examination using antibodies for SPINT1 and mitochondria or endosomes, as shown in Fig. 4B and Supplemental Figs. 2 and 3. A weak correlation between areas and DAB intensity (Fig. 4B). SPINT1 was not expressed in mitochondria and endosomes. However, SPINT1 was expressed in epithelial cells with enlarged cytoplasm rich in mitochondria, suggesting compensatory changes in response to epithelial degeneration.

Major comment (2):

SPINT1/HAI-1 is known to be expressed on the surface of epithelial cells, and immunohistochemical studies have demonstrated membranous immunoreactivity in normal epithelial cells. Therefore, the cytoplasmic expression lacking membranous immunoreactivity of SPINT1/HAI-1 in CP cells might be an interesting and novel finding. The authors should further investigate or at least discuss the biological significance of the cytoplasmic SPINT1/HAI-1 in CP cells.

To major comment (2): According to the reviewer’s comments, we performed double immunostaining using antibodies for SPINT1 and mitochondria or endosomes. As shown in Supplemental Figure 2, SPINT1 was not expressed in mitochondria and endosomes. On the other hand, Oberst et al. [16] (American Journal of Pathology, 2001, 158, 1301-1311) reported that the subcellular localization of the immunohistochemical staining for SPINT1/HAI-1 was observed in both the cytoplasm and at the cell membrane in human breast carcinoma cells. They described in the paper that the former may be explained by the internalization of the proteins or the synthetic pool of these molecules. In this study, SPINT1 immunoreactivity was localized at cytoplasmic membrane and the cytoplasm in the choroid plexus of mice, whereas that was localized in the cytoplasm in human aged brains. Accordingly, we added descriptions on cytoplasmic localization of SPINT1 in human brains in Discussion, based on the referenced paper [16]. The paper [16] was added in reference list. In addition, we additionally performed double immunohistochemical examinations using two kinds of antibodies for SPINT1 and mitochondria or endosomes. However, SPINT1 expression was not colocalized with mitochondria or endosomes. Accordingly, further studies are necessary to clarify localization of SPINT1 in the cytoplasm.

Minor comment:

Line 134: SPINT-1 should be SPINT1.

To the minor comment: Thank you for your comment. We spelled it wrong. We changed SPINT-1 to SPINT1.

Reviewer 2 Report

Comments and Suggestions for Authors

-A brief summary 

To analyze the function of the choroid plexus epithelium (CPE), the authors examined immunohistochemical expression of SPINT1, SIP1, and E-cadherin in the CPE, suggesting that they may be involved in transepithelial and junctional transport.

-Introduction

- The authors would like to finally mention the relationship between CPE function and neurodegenerative diseases, which seems to be the story of focusing on the expression of three molecules related to CPE function, but the current flow of the introduction reads as a gap between the sentences. Therefore, the introduction needs to be revised. Therefore, we request that the introduction be revised.

- Materials and methods 

- The authors collect human CP tissue from autopsy cases. We assume that CPE function declines with age, but it would be good to describe the rationale for thinking that there is no problem with this analysis in these cases.

- Regarding the methods of morphometrical analysis, the quantification of markers expressed in the cytoplasm should be evaluated in terms of the expression region and expression intensity. Please provide the basis for your opinion that the authors' analysis method is sufficient.

- Please explain the rationale for the use of two types of antibodies in the immunostaining of SPINT1.

- Results

- Regarding Figure 1. For each immunostaining, we would like to know if the staining was specific to CPE cytoplasm and not to the background or other structures of the CP.

- Regarding Fig. 1D, it is likely that the entire CP was used as the sample for RT-PCR. Also, there seems to be a shift in the size of the SPINT1 band. From these points, it is difficult to interpret this as the expression of SPINT1 in CPE. It would be better to revise the results so that the band sizes are at least the same. The full spelling of SI should be included in the figure legend.

- Regarding Figure 3, it says that 4608 cells were analyzed, but why was only one antibody, sc-137159, used for SPINT1? In the previous result, it is stated that there is no difference between the two types of monoclonal and polyclonal antibodies and the immunostaining results.

- Discussion

- The results of this paper, which show subcellular localization of SPINT1 and E-cadherin, seem to diverge from the results of CPE function and its relevance to neurodegenerative diseases caused by dysfunction of CPE. It would be better to add some additional results to provide evidence.

- Regarding the English text

- The English text seems to be different from what you want to say, so I think it would be better to proofread or correct the English text.

Author Response

To Reviewer 2:

Comment on -Introduction

(1)- The authors would like to finally mention the relationship between CPE function and neurodegenerative diseases, which seems to be the story of focusing on the expression of three molecules related to CPE function, but the current flow of the introduction reads as a gap between the sentences. Therefore, the introduction needs to be revised. Therefore, we request that the introduction be revised.

To the comment (1): Thank you for your useful comments. At present, the relationship between CPE dysfunction and neurodegenerative diseases is unclear. Accordingly, the introduction was revised according to the reviewer’s comments. We deleted 4 papers ([5-8] in the original version) for reference from the reference list and added a paper ([7] in the revised version) in the reference list.

Comment on - Materials and methods 

(2)- The authors collect human CP tissue from autopsy cases. We assume that CPE function declines with age, but it would be good to describe the rationale for thinking that there is no problem with this analysis in these cases.

To the comment (2): We first reported that immunoreactivity of glucose transporters 5 is in epithelial cells of the choroid plexus and ependymal cells, using autopsied 6 brains got from Kagawa University Hospital in 2014 (Neuroscience, 2014, 260, 149-157). Next, we reported immunohistochemical analysis of transporters related to clearance of amyloid-b peptides through blood-cerebrospinal fluid barrier in human brain, using autopsied 7 brains got from Kagawa University Hospital in 2015 (Histochemistry and Cell Biology, 2015, 144, 597-611). Third, we reported that immunoreactivity of glucose transporter 8 is localized in the epithelial cells of the choroid plexus and in ependymal cells, using autopsied 6 brains got from Kagawa University Hospital in 2016 (Histochemistry and Cell Biology, 2016, 146, 231-236). Fourth, we reported that immunoreactivity of urate transporters, GLUT9 and URAT1, is in epithelial cells of the choroid plexus of human brains, using autopsied 5 brains got from Kagawa University Hospital in 2017 (Neuroscience Letters, 2017, 659, 99-103). Fifth, we reported immunoreactivities for hepcidin, ferroportin, and hephaestin in astrocytes and choroid plexus epithelium of human brains, using autopsied 5 brains got from Kagawa University Hospital in 2020 [24] (Neuropathology, 2020, 40, 75-83). Sixth, we reported that sodium/glucose cotransporter 2 is expressed in choroid plexus epithelial cells and ependymal cells in human and mouse brains, using autopsied 6 brains got from Kagawa University Hospital in 2020 [19] (Neuropathology, 2020, 40, 482-491). Seventh, we reported immunoreactivity of receptor and transporters for lactate located in astrocytes and epithelial cells of choroid plexus of human brain, using autopsied 6 brains got from Kagawa University Hospital in 2021 (Neuroscience Letters, 2021, 741, 135479). Eighth, we reported immunohistochemical expression of osteopontin and collagens in choroid plexus of human brains, using autopsied 10 brains got from Kagawa University Hospital in 2022 (Wakamatsu K. et al., Neuropathology, 2020, 42, 117-125). Nineth, we reported that E-cadherin is expressed in epithelial cells of the choroid plexus in human and mouse brains, using autopsied 10 brains got from Kagawa University Hospital in 2023 (Current Issues in Molecular Biology, 2023, 45, 7813-7826).

Thus, we have continued to report the localization of several kinds of molecules including transporters and receptors using autopsied brains got from Kagawa University Hospital. Accordingly, we added a paper [20] (Chiba Y. et al., Int J Mol Sci, 2020, 21, 7230) to summarize these papers in the reference list. In addition, at autopsy, we removed brains from subjects as soon as possible after death. All brains were removed within 12 hours after death. We added data on the postmortem delay in Table 1. We confirmed immunoreactivity for SPINT1 in all ten brains. Accordingly, we considered that ten brain samples got from Kagawa University Hospital at autopsy were suitable for detection of molecules in epithelial cells including transporters and receptors.  We added descriptions to the rationale for thinking that there is no problem with this analysis in these cases, in Material and Methods.

(3)- Regarding the methods of morphometrical analysis, the quantification of markers expressed in the cytoplasm should be evaluated in terms of the expression region and expression intensity. Please provide the basis for your opinion that the authors' analysis method is sufficient.

To the comment (3): For morphometrical analysis, ten randomly selected, non-overlapping microscopic fields were analyzed from each section. In cases where the CP tissue area was insufficient to obtain ten unduplicated views, all available CP tissue regions within the section were analyzed. With the same method, we performed analysis of cell size of epithelial cells and reported results in a paper [18] (Murakami R. et al., Neuropathology, 2024). Concerning expression intensity, in the original version, we analyzed the ratio of immunoreactive area for SPINT1 to total segmented area, as the analysis was simple. The analysis of areas and brown rations revealed a weak positive correlation between them. However, the analysis did not take intensity into account. Accordingly, in the revised version, DAB intensity was quantified as the mean pixel value of the V-channel in the HSV color space (range: 0-255) and examined the correlation between areas of cells and DAB intensity. The findings indicated that the correlation was so weak (r = 0.298). Please look at Figure 4B in the revised version. In addition, the correlation between brown ratio and DAB intensity was strong (r = 0.836) (Supplemental Figure 3). These findings indicated increased immunoreactivity for SPINT1 in cells but did not precisely indicate increased expression for SPINT1 in epithelial cells with enlarged cytoplasm. Accordingly, a term “expression” was changed to “immunoreactivity” in Results.

(4)- Please explain the rationale for the use of two types of antibodies in the immunostaining of SPINT1.

To the comment (4): We deleted images for SPINT1 using polyclonal rabbit antibody for SPINT1 (GTX114793, Gene Tex) in human brains as the polyclonal antibody was slightly less specific rather than the mouse monoclonal antibody for SPINT1 (sc-137159, Santa Cruz) in human brains.

Comment on - Results

(5)- Regarding Figure 1. For each immunostaining, we would like to know if the staining was specific to CPE cytoplasm and not to the background or other structures of the CP.

To the comment (5): Please look at Figures 2 and 3 and Supplemental Figures 1 and 2. Immunoreactivity for SPINT1 was specific to CPE cytoplasm in human brains. No staining was seen in other structures in the choroid plexus, although immunoreactivity for SPINT1 was rarely seen in spindle cells in the choroid plexus of No. 9.

(6)- Regarding Fig. 1D, it is likely that the entire CP was used as the sample for RT-PCR. Also, there seems to be a shift in the size of the SPINT1 band. From these points, it is difficult to interpret this as the expression of SPINT1 in CPE. It would be better to revise the results so that the band sizes are at least the same. The full spelling of SI should be included in the figure legend.

To the comment (6): I apologize that Figure 1D in the original manuscript was misleading. The agarose gel was placed at an angle to the electric field of the electrophoresis tank, thus making the bands of the same molecular weight appear to be arranged at an angle. After the angle of the entire image was corrected, we confirmed that all the bands were located at the same molecular weight. Please look at Figure 1D in the revised version. We added the full spelling of SI.

(7)- Regarding Figure 3, it says that 4608 cells were analyzed, but why was only one antibody, sc-137159, used for SPINT1? In the previous result, it is stated that there is no difference between the two types of monoclonal and polyclonal antibodies and the immunostaining results.

To the comment (7): We deleted images for SPINT1 using polyclonal rabbit antibody for SPINT1 (GTX114793, Gene Tex) as the polyclonal antibody was slightly less specific rather than mouse monoclonal antibody for SPINT1 (sc-137159, Santa Cruz) in human brains.

Comment on - Discussion

(8)- The results of this paper, which show subcellular localization of SPINT1 and E-cadherin, seem to diverge from the results of CPE function and its relevance to neurodegenerative diseases caused by dysfunction of CPE. It would be better to add some additional results to provide evidence.

To the comment (8): Oberst et al. (American Journal of Pathology, 2001, 158, 1301-1311) [16] reported that the subcellular localization of the immunohistochemical staining for SPINT1/HAI-1 was observed in both the cytoplasm and at the cell membrane in human breast carcinoma cells. They described in the paper that the former may be explained by the internalization of the proteins or the synthetic pool of these molecules. We added descriptions on immunoreactivity for SPINT1 in the cytoplasm of human CP in Discussion. In addition, we examined double immunohistochemical investigations using antibodies for SPINT1 and mitochondria (Rockland, 909-301-D79) or endosomes (Novus, NBP1-30914). However, immunoreactivity for SPINT1 was not co-localized with that for mitochondria or endosomes. Accordingly, we added a paper for reference on endosomes [24] in the reference list. Further examinations were needed to identify specific subcellular locations.

Comment on - Regarding the English text

(9)- The English text seems to be different from what you want to say, so I think it would be better to proofread or correct the English text.

To the comment (9): According to the reviewer’s comment, further English editing was performed.

Round 2

Reviewer 1 Report

Comments and Suggestions for Authors

The authors responded adequately to my comments.  I have no additional comments on this manuscript.